# Endothelial Dysfunction in Huntington’s Disease: Pathophysiology and Therapeutic Implications

**DOI:** 10.3390/ijms26041432

**Published:** 2025-02-08

**Authors:** Ning Hu, Zihao Chen, Xinyue Zhao, Xin Peng, Yimeng Wu, Kai Yang, Taolei Sun

**Affiliations:** 1School of Chemistry, Chemical Engineering and Life Science, Wuhan University of Technology, 122 Luoshi Road, Wuhan 430070, China; 318526@whut.edu.cn (N.H.); 291734@whut.edu.cn (X.Z.); pengx000727@whut.edu.cn (X.P.); 332570@whut.edu.cn (Y.W.); 2Institute of WUT-AMU, Wuhan University of Technology, 122 Luoshi Road, Wuhan 430070, China; czh1120162494@126.com; 3Hubei Key Laboratory of Nanomedicine for Neurodegenerative Diseases, Wuhan University of Technology, Wuhan 430070, China; 4State Key Laboratory of Advanced Technology for Materials Synthesis and Processing, Wuhan University of Technology, 122 Luoshi Road, Wuhan 430070, China

**Keywords:** Huntington’s disease, endothelial dysfunction, angiogenesis, BBB, cerebral blood flow

## Abstract

Huntington’s disease (HD) is a progressive neurodegenerative disorder characterized by motor, cognitive, and psychiatric symptoms. While traditionally viewed through the lens of neuronal dysfunction, emerging evidence highlights the critical role of endothelial dysfunction in HD pathogenesis. This review provides a comprehensive overview of endothelial dysfunction in HD, drawing on findings from both animal models and human studies. Key features of endothelial dysfunction in HD include impaired angiogenesis, altered cerebral blood flow, compromised neurovascular coupling and cerebrovascular reactivity, and increased blood–brain barrier permeability. Genetic factors such as the mutant huntingtin protein, peroxisome proliferator-activated receptor gamma coactivator 1-alpha (PGC-1α), Brain-derived neurotrophic factor (BDNF), and the adenosine A2A receptor (ADORA2A) interact to influence endothelial function in complex ways. Various therapeutic approaches targeting endothelial dysfunction, including antioxidants, nitric oxide enhancers, calcium channel blockers, statins, and metformin, have shown promise in preclinical HD models but face translational challenges, particularly regarding optimal timing of intervention and patient stratification. The implications of these findings suggest that reconceptualizing HD as a neurovascular disorder, rather than purely neuronal, could lead to more effective treatment strategies. Future research priorities should include: (1) developing validated vascular biomarkers for disease progression, (2) advancing neuroimaging techniques to monitor endothelial dysfunction in real-time. These directions will be crucial for bridging the current gap between preclinical promise and clinical success in vascular-targeted HD therapeutics.

## 1. Introduction

HD is a hereditary neurodegenerative disorder primarily caused by a mutation in the huntingtin (HTT) gene, which results in an expanded cytosine-adenine-guanine (CAG) trinucleotide repeat. This mutation leads to the production of an aberrant form of the HTT protein, which is toxic to neurons and contributes to the progressive motor, cognitive, and psychiatric symptoms that characterize HD [1]. The hallmark of HD is the presence of involuntary movements, alongside cognitive impairments affecting executive function, attention, memory, and visuospatial skills, which profoundly impact daily living [1]. Psychiatric symptoms, including depression, anxiety, and psychosis, are prevalent and often precede the motor symptoms, highlighting the multifaceted nature of the disease [1].

Neuropathologically, HD is marked by the selective degeneration of neurons in the striatum and cortex, accompanied by a significant inflammatory response characterized by the proliferation of astrocytes and microglia [2,3]. At the cellular level, HD pathology is associated with disruptions in intracellular signaling, mitochondrial dysfunction, excitotoxicity, oxidative stress, and compromised protein clearance mechanisms [1]. The disease follows an autosomal dominant inheritance pattern, with the length of the CAG repeat expansion correlating with the age of onset and disease severity. Despite the absence of a cure, symptomatic treatments are available to manage the motor, cognitive, and psychiatric manifestations of HD, although the need for effective disease-modifying therapies remains critical [4].

Recent evidence indicates that the pathogenesis of HD extends beyond neuronal dysfunction, involving significant vascular contributions, particularly endothelial dysfunction. Endothelial cells, which line the inner surface of blood vessels, are vital for maintaining vascular homeostasis, regulating blood flow, and controlling the selective permeability of the blood–brain barrier (BBB) [5]. In HD, endothelial dysfunction is increasingly acknowledged as a central pathological feature, contributing to impaired BBB integrity, altered vascular tone, and dysregulated angiogenesis [6,7,8]. While the role of endothelial dysfunction has been extensively studied in other neurodegenerative conditions such as Alzheimer’s disease [9] and amyotrophic lateral sclerosis [10], its contribution to HD pathogenesis remains relatively underexplored. This is evidenced by recent comprehensive reviews of HD pathophysiology that primarily focus on neuronal mechanisms while giving limited attention to vascular aspects [1,11,12], and by the scarcity of clinical trials targeting vascular pathology in HD [13].

This review discusses the role of endothelial dysfunction in HD, focusing on five key areas: (1) impaired angiogenesis, (2) alterations in cerebral blood flow, (3) compromised cerebrovascular reactivity, (4) disrupted neurovascular coupling, and (5) BBB dysfunction. We also synthesize findings from both animal models and human studies, examine the influence of HD-associated genetic factors on endothelial function, and evaluate emerging therapeutic approaches. Additionally, we highlight critical gaps in current knowledge and propose future research directions that could advance our understanding of vascular contributions to HD pathogenesis.

## 2. Endothelial Dysfunction in HD

### 2.1. Impaired Angiogenesis

Recent investigations have revealed significant vascular alterations in HD-affected brains, particularly in regions most vulnerable to neurodegeneration. These modifications manifest as complex changes in blood vessel density, morphology, and functionality, suggesting a fundamental disruption of vascular homeostasis in HD pathogenesis [6,7,8]. While angiogenesis—the formation of new blood vessels—appears dysregulated in HD, the nature of this disruption is more complex than initially understood, involving both compensatory responses and pathological alterations [14].

A key feature of angiogenic dysfunction in HD is the dysregulation of vascular endothelial growth factor (VEGF) signaling. Elevated VEGF levels observed in both HD mice and patients [15] appear to serve a dual role: while initially representing a compensatory response to tissue hypoxia and metabolic stress, chronic VEGF elevation may paradoxically contribute to vascular dysfunction. This sustained VEGF upregulation often results in the formation of morphologically abnormal vessels with reduced diameter and compromised functionality [8]. The aberrant vessels, despite their increased density, exhibit impaired perfusion capacity and BBB integrity, potentially due to disrupted pericyte coverage and basement membrane composition [6].

The relationship between vessel diameter reduction and vascular function in HD is particularly significant. Decreased vessel diameter, observed consistently across multiple HD models and patient studies [15], has several critical implications for tissue perfusion and neuronal health:Reduced blood flow velocity and volume, leading to decreased oxygen and nutrient deliveryIncreased vascular resistance, potentially compromising the brain’s ability to regulate blood flow in response to metabolic demandsImpaired waste clearance, which may contribute to the accumulation of toxic metabolites

These vascular deficits show regional specificity that closely mirrors the pattern of neurodegeneration in HD. The striatum and cortex, regions primarily affected in HD, demonstrate the most pronounced vascular abnormalities [7]. This spatial correlation suggests a potential causative relationship between vascular dysfunction and neuronal loss, rather than merely representing a secondary consequence of neurodegeneration.

Brain-derived neurotrophic factor (BDNF), known to be decreased in HD patients’ serum [16], represents another critical link between angiogenic dysfunction and neurodegeneration. Beyond its well-established neurotrophic functions, BDNF plays a crucial role in promoting both angiogenesis and vessel stability. Its reduction in HD creates a detrimental cycle where impaired angiogenesis leads to reduced BDNF delivery, further exacerbating both vascular and neuronal dysfunction.

Studies using HD induced pluripotent stem cell (iPSC)-derived brain microvascular endothelial cells (iBMECs) have provided mechanistic insights into these angiogenic abnormalities. Enhanced WNT signaling activity in these cells drives increased but dysfunctional angiogenesis [17], suggesting that the problem lies not in the quantity but in the quality of newly formed vessels. This distinction is crucial for therapeutic targeting, as interventions may need to focus on normalizing vessel function rather than simply promoting angiogenesis.

The temporal relationship between vascular changes and disease progression is particularly noteworthy. Vascular abnormalities, including altered vessel density and morphology, are detectable in early symptomatic HD patients and even in pre-symptomatic stages in animal models [8]. This early manifestation suggests that vascular dysfunction may contribute to disease onset and progression, rather than merely representing a consequence of neurodegeneration. Furthermore, the presence of vascular changes in multiple brain regions, including the cortex, caudate/putamen, and substantia nigra, indicates a widespread impact on brain function that may explain the diverse symptomatology of HD (Figure 1) [8].

### 2.2. Altered Cerebral Blood Flow (CBF)

Early investigations into CBF in HD utilized computed tomography (CT) and single-photon emission computed tomography (SPECT) to identify hypoperfusion in both cortical and subcortical regions of the brain in HD patients [18,19,20,21,22]. While these pioneering studies were instrumental in establishing the potential link between impaired cerebral perfusion and neuronal death in HD, they had significant methodological limitations. SPECT imaging, despite being revolutionary for its time, offers relatively low spatial resolution (typically 8–10 mm) and relies on radioactive tracers that can introduce variability in measurements. Additionally, partial volume effects and limited temporal resolution in SPECT could mask subtle CBF changes, particularly in smaller subcortical structures critically affected in HD.

The research landscape was complicated by conflicting reports, highlighting the complexity of CBF alterations in HD. While some studies reported increased CBF [23], others found no significant changes [24]. These discrepancies likely stem from multiple factors beyond just imaging limitations. Patient heterogeneity, including variations in disease stage, CAG repeat length, and age of onset, could significantly influence CBF patterns. Moreover, regional variability in blood flow changes throughout disease progression may contribute to seemingly contradictory findings. The lack of standardized imaging protocols, including differences in data acquisition timing, processing methods, and reference region selection, further compounds these inconsistencies.

To overcome these methodological challenges, more advanced imaging techniques have revolutionized our understanding of CBF in HD. Arterial spin labeling (ASL) MRI has emerged as a superior tool, offering several advantages over traditional methods. Studies using continuous arterial spin labeling (CASL) [25] and pulsed arterial spin labeling (PASL) [26] have consistently reported reduced CBF in cortical and subcortical regions of both premanifest HD gene carriers and patients with manifest HD. The development of pseudo-continuous arterial spin labeling (pCASL) represents a significant advancement, combining CASL’s high signal-to-noise ratio with PASL’s reduced magnetization transfer effects. pCASL provides superior temporal resolution (typically < 5 s), enhanced sensitivity to regional flow differences, and improved quantification accuracy through longer labeling duration. Rocha et al. (2022) demonstrated the utility of pCASL by documenting significantly reduced CBF in the bilateral caudate and putamen regions of individuals with manifest HD compared to healthy controls [27], with spatial resolution reaching 2–3 mm and allowing precise delineation of subcortical structures.

The relationship between CBF changes and clinical symptoms in HD, particularly cognitive impairments, carries significant therapeutic implications. Early studies by Tanahashi et al. (1985) found that reduced CBF in hemispheric and frontotemporal regions correlated with lower scores on mini-mental status examinations [22], while Hasselbalch et al. (1992) [20] and Sax et al. (1996) [21] further established associations between decreased blood flow and cognitive deficits. These findings have opened new therapeutic avenues focusing on cerebrovascular health. Emerging strategies include the use of vasodilators to enhance regional blood flow, and anti-inflammatory agents to improve vascular function. The potential of these approaches is supported by Chen et al. (2012) [26], who demonstrated that decreased CBF correlated with poorer performance on the Stroop test. This suggests that therapies aimed at normalizing CBF patterns could help preserve cognitive function in HD patients. Additionally, CBF measurements using pCASL could serve as valuable biomarkers for early diagnosis and monitoring treatment efficacy, potentially enabling earlier therapeutic intervention before significant cognitive decline occurs (Figure 1).

### 2.3. Impaired Cerebrovascular Reactivity (CVR)

CVR is a crucial measure that reflects the brain’s ability to regulate blood flow in response to varying metabolic demands and perfusion pressures, which is vital for maintaining neuronal health. CVR is typically assessed by monitoring changes in CBF in response to vasoactive stimuli, such as increased levels of carbon dioxide (hypercapnia) or sensory stimulation [28]. This measure is particularly significant in understanding the dynamic interplay between CBF regulation and neuronal activity, as it ensures that active brain regions receive adequate oxygen and nutrients.

Recent studies in HD have provided insights into the impaired CVR in this condition, particularly through animal models and human studies. In HD R6/2 mice, impaired CVR to carbon dioxide (CO_2_) has been demonstrated using blood-oxygenation-level-dependent (BOLD) and flow-sensitive alternating inversion recovery (FAIR) MRI techniques. These investigations revealed a progressive decline in CVR in response to carbogen (a gas mixture of carbon dioxide and oxygen) in cortical and striatal regions of HD mice [15]. The observed impairment is potentially linked to a reduction in pericyte coverage of blood vessels, which plays a critical role in maintaining vascular function and integrity.

In human studies, impaired CVR has also been documented in subcortical white matter regions of HD patients, including the corpus callosum, forceps major, superior longitudinal fasciculus, and cingulum bundle [29]. These regions are integral to the brain’s structural and functional connectivity, and compromised CVR in these areas could contribute to the early neuropathological changes and clinical symptoms observed in HD, such as cognitive decline and motor dysfunction. The observation of impaired CVR in both cortical and subcortical regions highlights the widespread vascular dysregulation that occurs in HD and underscores the importance of further exploring CVR as a potential biomarker for disease progression and a target for therapeutic intervention (Figure 1).

### 2.4. Impaired Neurovascular Coupling (NVC)

NVC is a crucial physiological process by which the brain regulates CBF in response to changes in neuronal activity, ensuring that regions with heightened activity receive the necessary oxygen and nutrients to sustain optimal function. This process involves a complex interplay between neurons, glial cells, and cerebral blood vessels, mediated by various signaling molecules and cellular pathways. Recent evidence suggests that NVC disruption may serve as an early diagnostic marker in HD, with studies demonstrating functional alterations in neurovascular units before the onset of classical motor symptoms. Neuroimaging techniques, particularly functional MRI and advanced perfusion imaging, have revealed altered hemodynamic responses in pre-symptomatic HD gene carriers, suggesting that NVC assessment could potentially identify individuals at risk before clinical manifestation.

Research in HD models has significantly advanced our understanding of the mechanisms underlying NVC dysregulation, particularly regarding mitochondrial dysfunction’s central role. In the R6/1 HD mouse model, studies have shown age-dependent impairments in arterial contractility, which are closely associated with diminished mitochondrial support for vascular function [30]. The impact of mitochondrial dysfunction extends beyond local effects, creating a cascade of cellular stress that affects multiple aspects of vascular health. Compromised mitochondrial function leads to reduced ATP production, increased oxidative stress, and altered calcium handling, all of which contribute to endothelial dysfunction and impaired vascular tone regulation.

The R6/2 HD mouse model has provided insights into pre-symptomatic cardiovascular alterations, including enhanced endothelial-dependent dilation and impaired α1 adrenergic vasoconstrictor responses. This impaired vasoconstriction involves complex signaling pathways, particularly the dysregulation of nitric oxide (NO) signaling. The failure in recruiting NO-dependent pathways, coupled with increased endothelial-mediated relaxation, suggests fundamental alterations in vascular signaling mechanisms [31]. In addition, this study has also revealed significant changes in endothelial function within peripheral resistance arteries [31], emphasizing the systemic nature of endothelial dysfunction in HD. These findings suggest that endothelial dysfunction is not confined to the brain but also affects peripheral vasculature, potentially exacerbating the neurovascular and cognitive deficits observed in HD (Figure 1).

In summary, the interconnection between mitochondrial health and endothelial function represents a potential therapeutic target, with several promising strategies emerging as follows:Mitochondrial-targeted antioxidants: compounds designed to accumulate in mitochondria and reduce oxidative stress, potentially preserving both mitochondrial and endothelial function.NO pathway modulators: therapeutics targeting the NO signaling cascade to restore proper vascular tone regulation and improve neurovascular coupling.

These therapeutic strategies, particularly when implemented early in the disease course, may help preserve neurovascular function and potentially slow disease progression. The identification of NVC dysfunction as an early biomarker, combined with targeted interventions addressing both mitochondrial and vascular health, represents a promising avenue for future HD treatment approaches.

### 2.5. Increased Permeability of BBB

BBB plays a crucial role in maintaining central nervous system (CNS) homeostasis by selectively regulating the exchange of substances between the bloodstream and the brain. The BBB is composed of endothelial cells, astrocyte end-feet, pericytes, and tight junction proteins, which collectively ensure that essential nutrients, ions, and signaling molecules can enter the CNS while preventing the passage of large molecules, pathogens, and toxins [32,33]. The integrity of the BBB is vital for protecting neural tissue and supporting neuronal function, and disruptions to this barrier have been implicated in the pathogenesis of neurodegenerative diseases such as HD [32,33]. The assessment of BBB integrity in HD has historically relied on techniques such as the Evans blue (EB) permeability test, which has significant limitations including poor sensitivity to subtle changes, inability to detect region-specific alterations, and potential interference from tissue autofluorescence. More sophisticated approaches, such as dynamic contrast-enhanced magnetic resonance imaging (DCE-MRI), offer advantages including real-time monitoring, higher spatial resolution, and the ability to detect subtle BBB changes before overt dysfunction.

Initial studies using traditional BBB permeability assessment methods suggested that the BBB remains largely intact in HD mouse models, particularly in R6/2 mice [8,34]. However, more recent investigations employing advanced techniques have revealed significant BBB alterations in both genetic and chemical HD models. In early-stage R6/2 mice, researchers have detected BBB impairment through FITC-albumin leakage and dysregulation of tight junction proteins, indicating progressive deterioration of BBB integrity with disease progression [35]. These genetic models more closely mimic the gradual onset and progression of human HD, providing valuable insights into the temporal dynamics of BBB dysfunction in the disease process.

The use of chemical models, particularly those employing the mitochondrial toxin 3-nitropropanoic acid (3-NPA), has provided additional insights into the mechanisms of BBB dysfunction in HD. 3-NPA affects BBB integrity through multiple pathways, including direct oxidative damage to endothelial cells, disruption of the mitochondrial electron transport chain (ETC), increased reactive oxygen species (ROS) production, and ATP depletion leading to tight junction protein dysfunction. These mechanisms result in observable BBB disruption, manifested through the extravasation of blood components such as albumin and immunoglobulin G, complement factor infiltration, and enhanced EB dye penetration [34,36,37,38]. While 3-NPA models have been valuable in understanding the role of mitochondrial dysfunction in BBB disruption, their acute nature may not fully replicate the complex, progressive pathology seen in genetic HD.

Evidence from human HD studies has provided crucial validation of findings from animal models while revealing additional aspects of BBB dysfunction. Studies utilizing iBMECs from HD patients have demonstrated increased BBB permeability and reduced transendothelial electrical resistance (TEER), indicating structural and functional alterations at the BBB interface [17,39,40] (Figure 1). These human studies suggest more subtle and progressive changes compared to acute toxin models, highlighting the importance of considering disease stage and progression when interpreting BBB dysfunction in HD. The alignment between human and animal model findings confirms the relevance of BBB dysfunction in HD pathology while emphasizing the need for more sensitive detection methods.

The understanding of BBB dysfunction in HD has led to the development of several promising therapeutic strategies. These approaches include interventions targeting tight junction stabilization through occludin and claudin-targeting peptides, oxidative stress reduction via mitochondrial-targeted antioxidants and NOX inhibitors, and endothelial protection through agents promoting cell survival and mitochondrial function. Additionally, researchers are developing novel drug delivery strategies that account for altered BBB permeability and disease stage-specific changes. These therapeutic approaches must consider the progressive nature of BBB dysfunction in HD and may require different strategies at various disease stages.

## 3. Genetic Factors Influencing Endothelial Functions in HD

HD is primarily caused by an expanded CAG trinucleotide repeat in the HTT gene, which leads to the production of mHTT characterized by an elongated polyglutamine (polyQ) tract. This mutation is the primary genetic risk factor for HD, with the length of the CAG repeat inversely correlating with the age of symptom onset. Recent advancements in genetic research have uncovered several additional genetic modifiers, including PGC-1α [41], BDNF [42], and ADORA2A [43], all of which influence the risk and progression of HD. These modifiers not only affect endothelial functions individually but also interact in complex ways, producing synergistic or antagonistic effects that further complicate HD pathophysiology.

### 3.1. mHTT

Research has shown that mHTT significantly compromises BBB integrity through multiple mechanisms. In an isogenic juvenile iPSC model of HD, Linville et al. (2022) observed a significant reduction in endothelial cell adhesion—approximately two-fold—which correlated with decreased TEER and diminished expression of tight junction proteins [39]. This BBB disruption initiates a cascade of pathological events including increased neuroinflammatory signaling through NF-κB pathway activation, enhanced production of pro-inflammatory cytokines, infiltration of peripheral immune cells, and progressive deterioration of neurovascular unit function. Additionally, Lim et al. (2017) provided evidence that mHTT alters epigenetic markers critical for the regulation of gene expression related to barrier formation and maintenance, resulting in the downregulation of genes essential for tight junction integrity and endothelial cell adhesion [17].

Beyond BBB disruption, mHTT significantly impacts angiogenesis through multiple signaling pathways. Studies of iBMECs derived from iPSCs harboring 180 CAG repeats (HD180) demonstrate increased expression of VEGF receptor 2 (VEGFR2) and significantly enhanced endothelial sprout formation and branching in response to VEGF stimulation, indicating dysregulated angiogenic responses [39]. Transcriptomic analysis of HD iBMECs reveals altered angiogenic gene networks, particularly highlighting activation of the WNT/β-catenin pathway in human HD brain tissue [17] (Table 1). The WNT/β-catenin pathway plays a crucial role in the vascular pathology of HD, contributing to several key disruptions in vascular function. These include dysregulation of vessel stabilization, characterized by altered expression of vessel maturation factors, impaired pericyte recruitment, and compromised basement membrane assembly. Additionally, the pathway is implicated in impaired BBB development, evident through reduced expression of BBB-specific transporters, alterations in tight junction protein organization, and disrupted glucose transporter expression. Furthermore, angiogenic dysfunction in HD is marked by enhanced vessel sprouting, irregular vessel branching patterns, and compromised vessel stability. Notably, these WNT/β-catenin pathway alterations were shown to be reversible through pathway inhibition in vitro [17], suggesting potential therapeutic opportunities through WNT signaling modulation.

### 3.2. PGC-1α

PGC-1α emerges as a crucial modifier of HD pathology through its regulation of mitochondrial function and antioxidant defenses [44]. In endothelial cells, PGC-1α maintains vascular homeostasis through regulation of eNOS activity and NO production. Studies have shown that PGC-1α downregulation following Angiotensin II treatment correlates with decreased NO production and endothelial dysfunction, while its overexpression can counteract these effects through PI3K/Akt pathway activation [45,46]. Furthermore, PGC-1α plays a vital role in BBB integrity and angiogenesis, as demonstrated by studies showing reduced brain vascular permeability and enhanced tight junction protein expression following PGC-1α overexpression [47]. In pulmonary endothelial cells, PGC-1α deficiency leads to suppressed hypoxia-induced angiogenesis and exacerbated pulmonary hypertension [48] (Table 1).

### 3.3. BDNF

BDNF significantly impacts both neuronal and endothelial function through its interaction with the TrkB receptor, activating the PI3K/Akt pathway and leading to increased NO production [49]. Additionally, BDNF influences prostacyclin production, affecting vascular tone [50] (Table 1). The interaction between BDNF and other genetic modifiers creates a complex regulatory network where mHTT downregulates BDNF expression and transport [51], while PGC-1α promotes it [52]. This interplay demonstrates how genetic modifiers can work together to influence HD pathophysiology through both direct and indirect mechanisms affecting vascular function.

### 3.4. ADORA2A

ADORA2A exhibits context-dependent effects on vascular function in HD, with its role varying based on pathological conditions. In normal or inflammatory conditions, ADORA2A activation can protect vascular integrity by stabilizing tight junction proteins and reducing inflammatory signaling. However, under conditions of metabolic dysfunction, ADORA2A activity may contribute to pathological angiogenesis and increased vascular permeability. This dichotomy is evidenced by research showing that ADORA2A antagonism normalizes BBB permeability in insulin resistance [53], while its activation stabilizes endothelial tight junctions under inflammatory conditions [54]. Furthermore, ADORA2A’s role in angiogenesis adds complexity, as it promotes normal endothelial cell proliferation but may contribute to pathological angiogenesis through glycolytic pathway activation [54].

**Table 1 ijms-26-01432-t001:** Genetic factors influencing endothelial functions in HD.

Vascular Effect	Genetic Factor	Therapeutic Opportunities	References
Altered angiogenesis	mHTT	ASO, small molecule aggregation inhibitors	[17,39]
	PGC-1α	Mitochondrial enhancers, antioxidants	[48]
	ADORA2A	Selective agonists/antagonists	[54]
BBB disruption	mHTT	ASO, small molecule aggregation inhibitors	[17,39]
	PGC-1α	Mitochondrial enhancers, antioxidants	[47]
	ADORA2A	Selective agonists/antagonists	[53]
NO production	PGC-1α	Mitochondrial enhancers, antioxidants	[46]
	BDNF	TrkB agonists, BDNF mimetics	[49]

### 3.5. Integrated Genetic Factor Network in HD Vascular Pathology

The complex interplay between mHTT, PGC-1α, BDNF, and ADORA2A forms an intricate regulatory network that profoundly influences HD vascular pathology. These factors act through both primary mechanisms, such as mHTT-induced BBB disruption and PGC-1α-mediated mitochondrial responses, and through cross-regulatory interactions, including mHTT’s suppression of PGC-1α and BDNF expression. Understanding these interactions has important therapeutic implications, suggesting the potential benefit of combination therapies targeting multiple pathways simultaneously. Specific therapeutic opportunities include antisense oligonucleotides targeting mHTT, mitochondrial enhancers for PGC-1α-related dysfunction, TrkB agonists for BDNF pathway activation, and context-specific ADORA2A modulators.

## 4. Therapeutic Approaches Targeting Endothelial Functions in HD

Recent advances in understanding endothelial dysfunction in HD have led to the exploration of various therapeutic strategies. While preclinical studies have yielded promising results, the translation to effective human therapies has encountered significant challenges (Figure 2 and Table 2).

### 4.1. Antioxidants

Antioxidant interventions, including N-acetylcysteine (NAC), α-tocopherol, coenzyme Q10, and creatine, have shown considerable promise in animal models by reducing oxidative damage markers, delaying motor deficits, and improving survival rates [55,56,57,58,59,60].

However, translation to clinical success has been limited by several factors, including poor BBB penetration, insufficient tissue concentration, and suboptimal timing of interventions. The CREST-E trial’s failure to demonstrate significant benefits of creatine [61] and similar disappointing results with coenzyme Q10 [62] underscore the need for improved drug delivery systems and better understanding of therapeutic windows.

### 4.2. Enhancing NO Bioavailability

Strategies aimed at enhancing NO bioavailability have demonstrated potential, particularly through L-arginine supplementation and phosphodiesterase-5 (PDE-5) inhibition. Research has shown that increasing arginine concentrations can influence symptom onset and CBF in transgenic mice [63], while PDE5 inhibitors like sildenafil and vardenafil improved neurological outcomes in rat models [64]. The successful implementation of these approaches requires careful consideration of dosing parameters (L-arginine: 2–6 g/day; sildenafil: 50–100 mg daily) and timing of intervention, preferably during the pre-symptomatic phase. However, the long-term impact of chronic NO enhancement and potential systemic side effects necessitate careful monitoring and development of CNS-specific delivery methods.

### 4.3. Pharmacological Agents

Pharmacological approaches targeting multiple aspects of HD pathology have shown varying degrees of success. ACE inhibitors, particularly Trandolapril, have demonstrated neuroprotective effects in 3-NPA models by mitigating neurobehavioral deficits and restoring mitochondrial enzyme function [65], though adverse effects in some HD patients warrant careful consideration [66]. Calcium channel blockers have also shown promise, with Felodipine reducing mHTT aggregates in N171-82Q mice [67] and the peptide-based CTK 01512-2 improving motor function in BACHD mice [68,69].

Statins have emerged as particularly interesting candidates, demonstrating potential in delaying HD onset through multiple mechanisms including cholesterol modulation, anti-inflammatory effects, and enhanced endothelial function. Observational studies using the Enroll-HD dataset have suggested that statin therapy may delay symptom emergence in pre-motor HD patients [70].

### 4.4. Glycemic Control: Focus on Metformin

Metformin has emerged as a promising multi-target therapeutic approach, demonstrating varied benefits across different HD models. In transgenic mouse models, metformin extended survival time [71], reduced mHTT nuclear aggregation [72], and restored normal neuronal activity patterns [73]. The drug’s effects appear to work through multiple mechanisms, including AMPK activation and enhancement of lysosomal function. Implementation considerations include careful dose titration (starting at 500 mg daily, targeting 1000–2000 mg/day) and regular monitoring of renal function and vitamin B12 levels. The established safety profile of metformin makes it an attractive candidate for clinical translation, though optimal timing and duration of treatment remain to be determined.

In conclusion, the development of effective therapies for HD-related endothelial dysfunction requires addressing several key challenges. Future efforts should focus on developing targeted delivery systems to enhance BBB penetration, identifying patient-specific biomarkers for monitoring treatment efficacy, and optimizing combination therapy approaches. The varying efficacy of different interventions across disease stages suggests the need for stage-specific therapeutic strategies, potentially combining multiple approaches to address the complex pathology of HD. Success in translating promising preclinical findings will likely require a more nuanced understanding of drug delivery, timing of intervention, and patient-specific factors that influence treatment response.

## 5. Possible Mechanism of Endothelial Dysfunctions in HD

Recent advances in understanding metabolic dysfunction in HD have revealed striking parallels with other conditions characterized by vascular compromise, particularly in the context of the Warburg effect and its downstream consequences. In HD, mHTT triggers a cascade of metabolic alterations that prominently features increased reliance on aerobic glycolysis, mirroring the metabolic reprogramming observed in various pathological conditions [74]. This metabolic shift is mechanistically linked to the upregulation of the WNT/β-catenin signaling pathway, which orchestrates the expression of key metabolic enzymes including glucose transporters, pyruvate kinase M2 (PKM2), and lactate dehydrogenase A (LDH-A), while simultaneously suppressing the pyruvate dehydrogenase complex (PDH) [74].

The consequences of this metabolic reorganization extend beyond simple energy metabolism. Of particular significance is the accumulation of lactate, which has emerged as a critical mediator of vascular dysfunction through previously unrecognized mechanisms. Drawing insights from recent studies in sepsis-induced acute lung injury, lactate has been shown to promote histone lactylation and accelerate glycocalyx degradation, ultimately compromising vascular integrity [75]. These findings have important implications for HD, where chronic inflammation and metabolic disruption may operate through similar pathways. The impairment of mitochondrial function by mHTT forces cells to increase their reliance on glycolysis, leading to elevated lactate production [75]. This metabolic adaptation, while initially protective, may ultimately contribute to disease progression through effects on vascular function and BBB integrity. The convergence of these pathways suggests novel therapeutic opportunities. Targeting lactate metabolism and protecting glycocalyx integrity could provide innovative approaches to mitigating vascular dysfunction in HD. This is particularly relevant given the observed elevation in lactate/pyruvate ratios in HD models, indicating sustained metabolic stress [76].

## 6. Conclusions

This comprehensive review highlights the critical role of endothelial dysfunction in HD pathophysiology, offering new insights into disease mechanisms and potential therapeutic strategies. Our analysis underscores several key findings that deepen the understanding of HD pathogenesis. The multifaceted nature of endothelial dysfunction in HD is characterized by impaired angiogenesis, altered CBF, compromised NVC, and increased BBB permeability. These vascular abnormalities may not only precede but also exacerbate the neurodegenerative processes typically associated with HD. The complex interplay between genetic factors, including mHTT, PGC-1α, BDNF, and ADORA2A, demonstrates the intricate relationship between genetic predisposition and vascular pathology, necessitating a more holistic approach to HD treatment. While preclinical studies targeting endothelial function show promise, their translation into effective clinical therapies remains challenging, emphasizing the need for refined translational strategies and personalized approaches addressing both neuronal and vascular components of the disease.

The findings of this review contribute to a paradigm shift in HD research, expanding focus beyond neuronal mechanisms to include vascular factors. This broader perspective identifies specific vascular biomarkers for monitoring disease progression and treatment efficacy, including quantitative CBF metrics through ASL and dynamic susceptibility contrast (DSC) imaging, BBB integrity measures such as DCE-MRI and CSF/serum albumin ratios, and endothelial function indicators including circulating progenitor cells (CPCs) and plasma markers of endothelial activation. Future research priorities should focus on developing advanced neuroimaging protocols for early detection, implementing longitudinal clinical trials that incorporate vascular biomarkers, and exploring emerging therapeutic approaches such as cell-based therapies, novel drug delivery systems, and gut microbiome–brain axis modulation. Particular emphasis should be placed on studying early-stage HD patients and pre-symptomatic gene carriers to identify early vascular changes and evaluate preventive interventions.

Recognition of endothelial dysfunction as a central component of HD pathology opens new avenues for therapeutic intervention and disease monitoring. A two-pronged approach to therapy development is suggested, combining early intervention strategies focused on vascular protection in pre-symptomatic stages with disease modification approaches targeting stage-specific therapeutic combinations. The integration of vascular biomarkers into clinical practice and research protocols will be crucial for advancing our understanding of HD progression and developing more effective treatments. Success in this endeavor will require coordinated efforts to validate biomarkers, standardize imaging protocols, and develop targeted therapeutic approaches that address both the vascular and neuronal aspects of this devastating neurodegenerative disorder (Figure 3). Future clinical trials should be designed to evaluate combination therapies that simultaneously target vascular health and neuronal survival, with careful attention to timing of intervention and patient-specific factors that may influence treatment response.

## Figures and Tables

**Figure 1 ijms-26-01432-f001:**
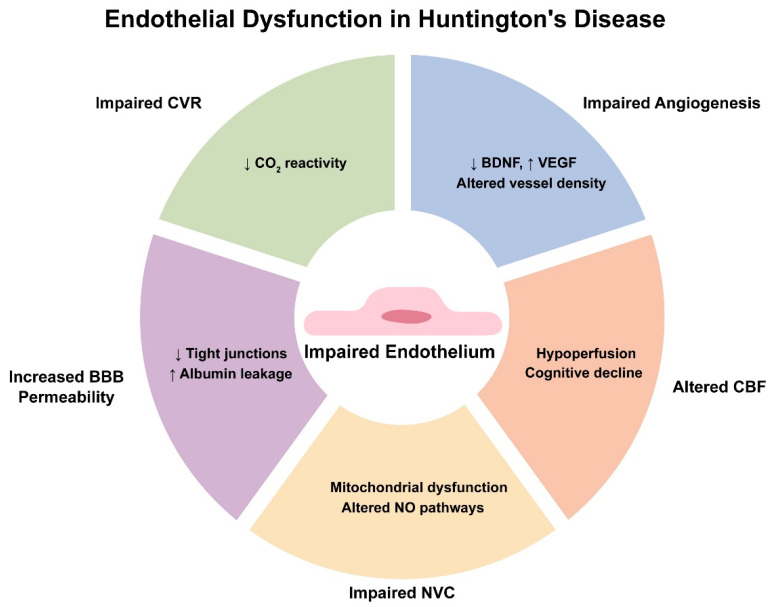
Key Features of Endothelial Dysfunction in HD. It summarizes the major aspects of endothelial dysfunction observed in HD, including impaired angiogenesis, altered CBF, impaired NVC, increased BBB Permeability and impaired CVR. These interconnected factors collectively contribute to the complex vascular pathology observed in HD, potentially playing a significant role in disease progression and symptom manifestation.

**Figure 2 ijms-26-01432-f002:**
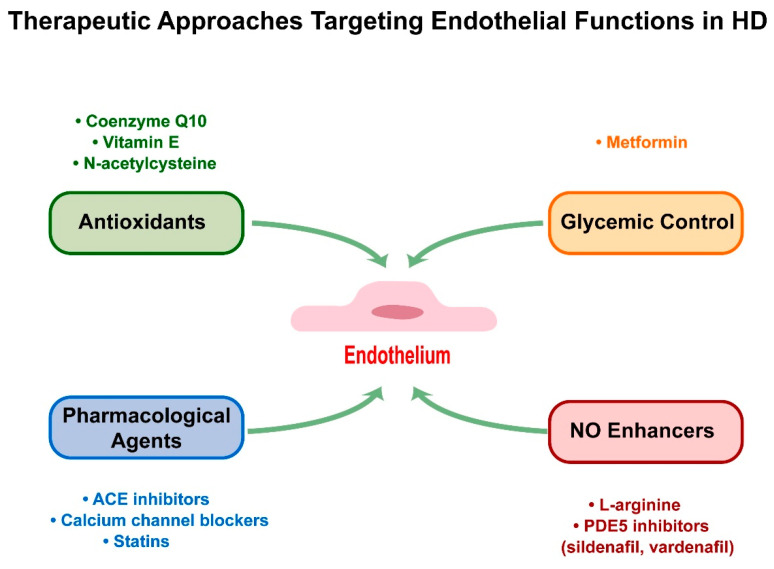
Therapeutic approaches targeting endothelial functions in HD. This figure illustrates the key therapeutic strategies aimed at improving endothelial function in HD, including antioxidants, NO enhancers, pharmacological agents, and glycemic control. These diverse approaches address different aspects of endothelial dysfunction in HD, offering potential avenues for therapeutic intervention and disease management.

**Figure 3 ijms-26-01432-f003:**
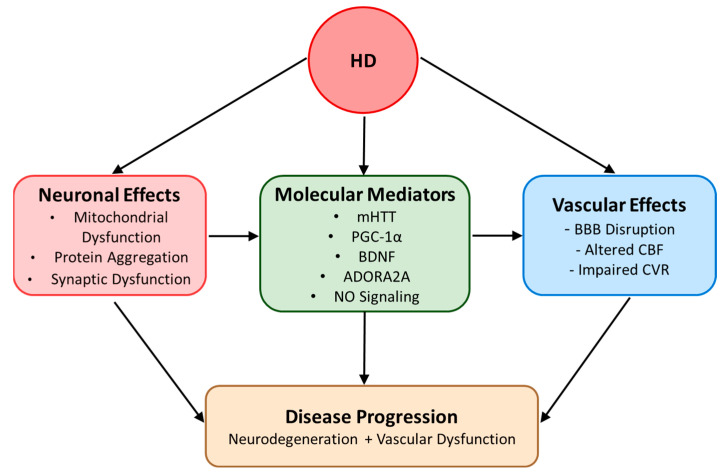
The interplay between neuronal and vascular pathology in HD. This figure illustrates the multifaceted pathophysiology of HD, emphasizing the interconnected nature of neuronal and vascular dysfunction. Primary neuronal effects (red) include mitochondrial dysfunction, protein aggregation, and synaptic dysfunction. The vascular effects (blue) are characterized by BBB disruption, altered CBF, and impaired CVR. Together, these pathological processes contribute to disease progression (orange) through combined neurodegeneration and vascular dysfunction, highlighting the importance of targeting both neuronal and vascular pathways in therapeutic interventions.

**Table 2 ijms-26-01432-t002:** The strengths and limitations of each therapeutic strategy in HD.

Approach	Key Advantages	Limitations	Optimization Strategies
Antioxidants	Multiple targets, well-tolerated	Poor BBB penetration, timing issues	Novel delivery systems, early intervention
NO Enhancers	Direct vascular effects	Systemic side effects	CNS-targeted delivery, biomarker monitoring
ACE Inhibitors	Proven safety profile	Variable neurological effects	Patient-specific titration
CCBs	Strong preclinical evidence	Cardiovascular effects	Novel formulations, targeted delivery
Statins	Multiple mechanisms	Muscle-related side effects	Combination therapy, dose optimization
Metformin	Multi-target effects	Variable response	Personalized dosing, early intervention

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
