# Peer review of "Endothelial Dysfunction in Huntington’s Disease: Pathophysiology and Therapeutic Implications"

_ijms, 2025, doi:10.3390/ijms26041432_

Round 1
Reviewer 1 Report
Comments and Suggestions for Authors
The proposals for supplementing the manuscript are described in the attached report.

Reviewer 2 Report
Comments and Suggestions for Authors
1. please add references for the following line, 42, 58, 60
2. line 100-102, please add refs
3. line259 seems incomplete "sprout density"
4. table 1 and 2 can you maintain consistency with references style ex:instead of Hu et al, put[1]
5. 4.3 talks about several pharmacological agents are these agents 4.4 4.5 etc..?
if so instead of labeling them as separate sections label as 4.3.1.,4.3.2. this will avoid confusion.
6. line 370, 373, 383 add references for the statements.
7. 4.8 is conclusion and 5 is conclusion, can you rephrase this section?
8. please delete the comments for acknowledgement and funding
9. 3.2 in red, please correct
